# Using Sound Location to Monitor Farrowing in Sows

**DOI:** 10.3390/ani13223538

**Published:** 2023-11-16

**Authors:** Elaine van Erp-van der Kooij, Lois F. de Graaf, Dennis A. de Kruijff, Daphne Pellegrom, Renilda de Rooij, Nian I. T. Welters, Jeroen van Poppel

**Affiliations:** 1Department of Animal Husbandry, HAS Green Academy, University of Applied Sciences, P.O. Box 90108, 5200 MA ‘s-Hertogenbosch, The Netherlandsj.vanpoppel@has.nl (J.v.P.); 2Department of Applied Biology, HAS Green Academy, University of Applied Sciences, P.O. Box 90108, 5200 MA ‘s-Hertogenbosch, The Netherlands

**Keywords:** pigs, animal welfare, sound camera, behaviour, piglet crushing

## Abstract

**Simple Summary:**

Automated monitoring can help farmers care during the farrowing of sows and piglets. Five sows were monitored in two field studies. A sound camera with small microphones showing sounds as coloured spots and a security camera was used to record the farrowing of sows and piglets. First, sound spots were compared with audible sounds, analysing video data. This gave many false positives, including visible sound spots but no audible sounds. Of piglet births, 23 of 50 piglet births were visible, but none were audible. The sow’s behaviour changed when farrowing started. One piglet was silently crushed. Secondly, data were analysed at a slower speed, and sound spots were compared with sounds and behaviour separately. This resulted in better results, but again, many sound spots showed without audible sound. When adding up audible sounds and visible sow behaviour and comparing sound spots with the combination of sound and behaviour, results were much improved, with an accuracy of 91.2%, an error of 8.8%, a sensitivity of 99.6%, and a specificity of 69.7%. We conclude that sound cameras are promising tools, detecting sounds more accurately than humans. The most promising application for the sound camera is detecting the onset of farrowing.

**Abstract:**

Precision Livestock Farming systems can help pig farmers prevent health and welfare issues around farrowing. Five sows were monitored in two field studies. A Sorama L642V sound camera, visualising sound sources as coloured spots using a 64-microphone array, and a Bascom XD10-4 security camera with a built-in microphone were used in a farrowing unit. Firstly, sound spots were compared with audible sounds, using the Observer XT (Noldus Information Technology), analysing video data at normal speed. This gave many false positives, including visible sound spots without audible sounds. In total, 23 of 50 piglet births were visible, but none were audible. The sow’s behaviour changed when farrowing started. One piglet was silently crushed. Secondly, data were analysed at a 10-fold slower speed when comparing sound spots with audible sounds and sow behaviour. This improved results, but accuracy and specificity were still low. When combining audible sound with visible sow behaviour and comparing sound spots with combined sound and behaviour, the accuracy was 91.2%, the error was 8.8%, the sensitivity was 99.6%, and the specificity was 69.7%. We conclude that sound cameras are promising tools, detecting sound more accurately than the human ear. There is potential to use sound cameras to detect the onset of farrowing, but more research is needed to detect piglet births or crushing.

## 1. Introduction

It is estimated that by 2050, the human world population could reach >9 billion, consuming 50–60% more food than at present. The majority of people still prefer animal proteins over plant-based food, and the demand for livestock products continues to grow, as does global food insecurity [1]. Sustainable intensification is one of the solutions [2]. With the intensification of food production and the industrialisation of animal production systems comes the fear of decreased animal welfare [3]. While meat production increases, society expects that animals used for meat are treated humanely and individually. Precision Livestock Farming (PLF) can improve or monitor animal welfare on farms if properly implemented [4,5]. PLF can be defined as managing livestock production using the principles of process engineering. Smart sensors are used to measure and monitor animal health and welfare [6,7,8,9,10]. Several sensors have been developed for the livestock sector. For pigs, the main focus is on the health and productivity of pigs, with sensors such as cameras, microphones, thermometers, and accelerometers being developed and applied [1,11,12,13,14]. A new development is the application of a sound camera, providing sound source localisation through an array of 64 microphones and visualising sound sources as coloured spots. Sound cameras are presently used for crowd control under outdoor conditions [15] and have been introduced in ecology [16] and agriculture, specifically in poultry and pigs [17,18,19]. Potentially, these cameras can be of use in monitoring welfare in pig farms since automated behaviour monitoring via sound and vision could help farmers to prevent health and welfare issues [20,21,22,23,24]. 

The farrowing phase in pig production and breeding is a crucial moment that has a great impact on the economics of the farm but also on the welfare of the sow and piglets. Piglet mortality is a big problem in modern sow farming [25]. For sow farms, relevant welfare issues around farrowing are a prolonged birthing process, possibly resulting in stillborn piglets and the crushing of piglets after farrowing [26,27,28]. Globally, approximately one piglet per litter is stillborn [29]. More stillbirths occur in sows that are confined in a crate during farrowing than in sows in open pens [30]. Stillbirths are often due to asphyxiation during the farrowing process. Piglets born later in the farrowing process have a higher chance of asphyxiation [29]. Asphyxiation is related to dystocia in sows due to prolonged farrowing or weak uterine contractions. This requires the stockman to assist the sow during farrowing. Improved farrowing management and human supervision during farrowing might decrease piglet mortality [25]. The automated monitoring of sows and alerting the farmer when the farrowing process stagnates can aid the farmer in optimising the health and welfare of his animals [8]. Therefore, in this study, we focus on the automatic monitoring of the farrowing process. Sows show specific behaviours and behavioural patterns before, during, and after farrowing. If we can monitor these behaviours, we might detect the onset of farrowing and follow the process of farrowing. Before farrowing, sows show a natural pattern of nest-building behaviour, such as foraging, rooting, and pawing, which are motivated by the desire to build a shelter to protect their offspring against predators and cold weather. Domestic sows, when given nest-building materials, still perform nest-building behaviours [31]. In farming systems with farrowing crates, nest-building behaviour is reflected in ‘playing’ with the available material, which can be a jute sack or a handful of straw. During farrowing, sows show pain-related behaviour such as tail flicks, straining, and pushing the back leg forward [30]. Approximately 50% of postnatal deaths in piglets are caused by the crushing of the sow [28]. Piglet crushing happens mostly within the first four days after farrowing. There is a large individual variation in piglet crushing between sows; sows show a more protective mothering style and crush fewer piglets. Especially during posture changes of the sows, the young piglets are at risk. Piglets vocalise during trapping events but also during other stress-related events. However, detecting these sounds might be used in a monitoring device [32]. 

In this study, we have used a sound camera together with a security camera to monitor sounds, vision, and sound location around farrowing as the first step in developing a sound-based early warning system for a stagnating birthing process and the prevention of piglet crushing.

## 2. Materials and Methods

### 2.1. Sound Camera

The L642V is a camera with an array of 64 microphones. The device uses delay-and-sum beamforming to localise different sound sources and visualises these on an acoustic map. Delay-and-sum beamforming signal processing can be divided into four steps. Since the sound of each source travels to every microphone along a different path (Step 1), the signals captured by the microphones are similar in wave form but differ in their delay and phase. Delay and phase are proportional to the travelled distances. The delays can be determined from the sound speed, the distance between the microphones, and the sound sources (Step 2). The Beamformer targets the point of one of the sound sources, shifting the signal of each microphone via the difference in runtime depending on the focus point. Therefore, the signal components of this one sound source are in phase, and of the other sound sources are out of phase (Step 3). Finally, the signals of the microphones are summed together and normalised by the number of microphones (Step 4). If a certain target point does not contain a source, the signals partly cancel each other out due to destructive interference. At target points with a source, the signals align, and add up due to positive interference. The maximum amplitude is calculated from the time signal and the sound source is visualised on the acoustic map. Due to the positive interference, target points with a source have a higher magnitude than those that do not, and thus, source locations can be found. In this study, we denoised sound by selecting a frequency band for which the sounds were visualised. In a pilot study, we tested different frequency bands and found that in a range from 39,000 to 49,000 Hz, the sounds of the sow and her environment were best visible, with the lowest influence of background noise. In the present study, each camera was manually tuned to a specific optimal frequency band of 2000 Hz within these limits, according to the noise in the farrowing unit in that period. The camera has a built-in spatial filter, which means it only shows sound sources within the selected area, which, in our study, was the farrowing pen for each sow. Finally, the camera also denoised automatically by not visualising a sound when no clear source could be found. 

### 2.2. Study Design

The study was performed at a commercial pig farm with two farrowing units for 64 sows each. Sows were housed in farrowing crates within a farrowing pen, with access to a jute sack in the pen but no straw and a solid concrete floor. Sows were monitored around farrowing, staying in a farrowing pen of 2.80 × 1.75 m with a farrowing crate of 2.1 × 1.0 m. Two Bascom XD10-4 security cameras that showed sound and vision were placed above the pens to record audible sounds and the behaviour of the sow and piglets, with each camera viewing two pens. Three Sorama L642V sound cameras were placed directly above three farrowing pens, with each camera viewing one pen (Figure 1). 

Data from the cameras could not directly be recorded due to the safety settings. We, therefore, streamed the data to three laptops in the office of the farm. The screen of the laptops showed the image of the security camera and the sound camera side by side, as well as a clock, to synchronise the images if necessary (Figure 2). We recorded data using the screen recorder software Open Broadcast Software version 27.2.1 (OBS-studio), resulting in video files in the mp3 format. Laptops were remotely controlled using TeamViewer.

Approximately 45 min of video data from each of the four sows were analysed after the first field study. Birthing events and crushing events were recorded from the video footage using the Observer XT (Noldus Information Technology). Four hours before farrowing and during farrowing, sow behaviours were recorded that were possibly associated with the birthing process. We used a simplified ethogram with three behavioural categories: lying (inactive), playing with the jute sack/rooting (snout on the concrete floor), and sitting/standing (inactive). Audible sounds were recorded from the recorded video of the security camera. Visible sounds were recorded from the sound camera. When a sound spot was visible at roughly the right location within ±1 s from the audible sound, the spot was considered correct and positive. Audible sounds with no corresponding sound spot were considered false negatives. Sound spots with no audible sounds were considered false positives. Finally, when no sound was visible or audible for 2 s, this was considered correct negative. In Table 1, the connected audible and visible sounds and sound locations are shown.

In Table 2, the connected audible and visible sounds and behaviours are shown.

Data from three sows were analysed after the second field study. From the security cameras, sow behaviours, and audible sounds were recorded using the Observer XT, and from the sound camera, sound spots were recorded for a short period during farrowing. When a sound spot was visible at roughly the right location within ±1 s from the audible sound, the spot was considered correct and positive (CP). Audible sounds with no corresponding sound spot were considered false negatives (FN). Sound spots with no audible sounds were considered false positives (FP). Finally, when no sound was visible or audible for 2 s, this was considered correct negative (CN). For validation purposes, accuracy, error%, sensitivity and specificity were calculated as follows:$$accuracy=(\frac{\left(CP+CN\right)}{total}\;)\times 100\%$$
$$error\;\%=(\frac{\left(FP+FN\right)}{total}\;)\times 100\%$$
$$sensitivity=\left(\frac{CP}{FN+CP}\right)\times 100\%$$
$$specificity=(\frac{CN}{\left(FN+CP\right)}\;)\times 100\%$$

## 3. Results

Data from five sows in two field studies were gathered and analysed for visible sounds (sound spots) and audible sounds, and, in field test 2, visible sow behaviour as well. There was a minor time lag in the recording of the visible sounds of approximately 1.5 s, which was corrected by adding 1.5 s to the recorded times of the sound spots. We adjusted the frequency settings for each camera per sow manually by testing which frequency band showed the best visualising of sound sources with the least noise at that time and place. This resulted in frequency bands of 39,570–41,570 Hz (camera 1), 46,060–48,060 Hz (camera 2) and 41,730–43,730 Hz (camera 3) in the first study and an adjustment to 45,630–47,630 Hz for camera 1 in the second study. 

### 3.1. Field Study 1

In the first field study, we compared the audible and visible sounds of the sows before farrowing and recorded 13,351 sound spots and 981 audible sounds in 177 min of video data. We found a low agreement between the sound and vision data (Table 3). 

### 3.2. Field Study 2

In the second field study, we analysed 6 min (360 s) of video from two sows during farrowing. Video data were analysed at a 10-fold slower speed, and audible sounds, sow behaviour, and sound spots were recorded. This resulted in a somewhat higher but still unsatisfactory agreement between the sound and vision data when sound spots were compared to either audible sounds or visible behaviour (Table 4). However, when comparing sound spots with the combination of audible sounds and visible behaviour, results were much improved, with an accuracy of 91.2, an error% of 8.8, and a specificity of 69.7 (Table 4). In this analysis, we added up sounds and behaviours to compare with sound spots, considering an event as a correct positive if either sound or behaviour (or both) were shown at the same time and location as the sound spot.

### 3.3. Birthing Events and Piglet Crushing

For the detection of birthing events, data from four sows were used. A total of 50 piglets were born from these sows. For 23 of 50 piglet births, a sound spot was visible at the correct time; piglet births were usually not audible, but most were visible on the safety camera. This resulted in an accuracy of 71.4%. Sound spots were visible in the area behind the sow for some time after the piglet was born, but the movement of the piglets was not audible. 

One piglet was crushed shortly after birth, but no sound was heard, and no sound spots were visible for this event, which was only visible on the video footage of the Bascom camera. 

When we analysed behaviours around parturition, we found that the sows showed specific behaviours at specific times: playing with the jute sack in the farrowing pen was seen more before parturition and especially during the last hours before the piglets were born. Once farrowing started, this behaviour stopped almost completely (Table 5). During farrowing, typical movements of the legs of the sows were seen, seemingly associated with the birthing events. 

## 4. Discussion

For this study, we used data from five sows to study the application of a sound camera in pig farming. Further research is needed using more sows and more repetitions to validate these findings. However, in this study, we were able to optimise the methods for analysing the results of sound spots, sound, and behaviour, and we found promising results for the application of the sound camera to monitor farrowing sows. 

In this study, the human observer was the gold standard for audible sound. When comparing manual to automated scoring, there were some problems with finding the gold standard. Clinical research has shown that manual scores are usually qualitative or semiquantitative and subjective, even when conducted by a seasoned observer, while automated image analysis is quantitative, reproducible, and objective. Manual image analysis has some drawbacks [33] that can easily be extrapolated to manual sound analysis. The sources of bias include the illusion of size (size being influenced by the context), distinguishing colours, and lateral inhibition (increased response to edges). For sound analysis, these can translate into an illusion of loudness (being influenced by the loudness of other sounds), a distinguishing pitch (depending on the pitch of surrounding sounds), and an increased response to short and sharply defined sounds. General sources of bias include inattentional blindness (i.e., not paying attention) and confirmation bias (i.e., hearing what you expect or want to hear). Labelling audible sounds from videos recorded with a safety camera probably resulted in many false negatives for audible sounds and inaccuracies in labelling since the human observer either hears the sound and reacts too late or does not hear the sound at all, while the sound camera does receive the sound. Furthermore, the labelling of the sound spots was probably not accurate enough since we labelled at a normal speed. This resulted in many ‘cloud of sound spots’ events, with a cluster of sound spots occurring at once. Playing the videos at a 10-fold slower speed showed that the sound spot clouds were actually a series of sound spots that started with one spot in the correct place, followed by a cluster of spots in the area. Therefore, we adjusted the analysis for the second field study. 

Most sows farrowed at night, with low visibility on the cameras, which increased the number of false positives (i.e., sounds visible in a different spot than audible) due to human error. In all tests, we counted many more sound spots than visible behaviours or audible sounds. This may very well be due to human error. A reliability analysis for labelling sound spots between the observers showed an agreement of 82%, but a Kappa value of 0.17 (indicating slight agreement) was obtained. The high number of sound spots and almost no silent periods lead to a high agreement by a chance of 0.78. This lowered the Kappa value [34,35]. In addition, the manual labelling of data as the gold standard is a point of discussion.

In the second field study, adding visible behaviour gave much better results, correctly classifying sound spots for visible but inaudible behaviours. Furthermore, we recorded sound from the Bascon camera and listened to the recordings, which is an indirect way of working with data and probably lowered the audibility for human observers. It seems that the sound camera is much more accurate and precise at detecting sounds than we, as human observers, are. Therefore, we advise combining audible sounds and visible behaviours when validating sound location sensors such as the sound camera we used. 

We detected less than half of the piglet birth events with the sound camera. These sounds were low-pitched and soft. Filtering the noise from the sound camera and using the high-frequency bands to visualise sounds might have increased the number of false negatives. One piglet was crushed during the study without an audible sound or a sound spot from the sound camera. We expected to capture high-pitched sounds that accompanied crushing events, but if it happened fast, the piglet had no time to scream. From only one crushing event, we could not validate whether the sound camera could detect piglet crushing. In a study where sound was used to detect piglet distress, it was found that many more piglet stress calls were associated with other stress-related events than associated with trapping events. Although adding context-based event filters increased the results, sound might not be the preferred method for detecting crushing events [32]. Crushing events are mostly associated with posture changes in the sow, such as rolling over during resting. Sows that crush fewer piglets show fewer posture changes [28]. If these posture changes could be detected with a sound camera, this might be used to prevent the crushing of piglets. 

The automatic monitoring of sow behaviours can be performed using cameras or activity meters [20,36,37,38]. To investigate whether the sound camera can be used to detect specific behaviours associated with the onset of farrowing, behaviours were recorded around farrowing. We found that playing with the jute sack and rooting was seen almost only before farrowing. This behaviour is associated with nest-building behaviour [31]. Typical leg movements were seen predominantly during farrowing and seemed to be associated with birthing events. These behaviours resembled the leg behaviours that were reported in previous research, which were classified as pain-related behaviours during farrowing [30,39]. Nest building behaviour was only seen before farrowing, and leg movements during farrowing; after farrowing, these behaviours stopped completely. These behaviours were associated with sound spots in the corresponding, typical locations as follows: playing with the jute sack or rooting on the floor was seen as sound spots near the head of the sow, and leg movements were seen as sound spots near the front or hind legs, detected with the sound camera. This is interesting because the cross-over from showing nest-building behaviour (playing with the jute sack and rooting on the floor) to lying down and showing only some typical leg movements seems to mark the onset of farrowing. If so, we could use the detection of these behaviours with the camera as a signal for the farmer that farrowing has started. Furthermore, the absence of both these behaviours, with sows lying down and being fully inactive, marks the end of the farrowing process. This could be combined with the fact that sound spots behind the sow showed the movement of piglets that were just being born. As long as new spots appeared behind the sow, the farrowing process was still in progress. The combination of these findings could be used to monitor the farrowing process and its duration. However, more research is needed to validate these assumptions. 

## 5. Conclusions

Sound cameras are potentially interesting to apply in pig farming since they can detect sounds and sound locations better than the human observer. The behaviour of the sow and the movement of piglets in the pen could be reliably detected. However, we could not reliably detect piglet births and crushing events in this study. When analysing sound and visual data, it is important that a slower speed must be used to record the order of events and sound spots and that sound data are connected not only to audible sounds but also to behaviours that are inaudible to the human ear. The most interesting application for sound cameras seems to be detecting the onset of farrowing by recording sounds from the prepartum sow as she is preparing for the farrowing process and monitoring the farrowing process by detecting sounds of newly born piglets in the area behind the sow. Further research is needed to test this application, using more sows and more repetitions. 

## Figures and Tables

**Figure 1 animals-13-03538-f001:**
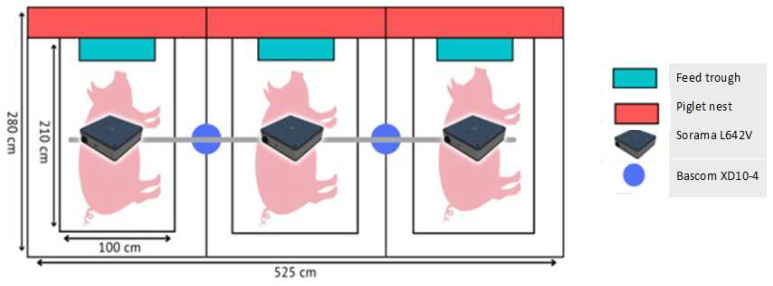
Experimental set-up for three sows with two Bascom XD10-4 security cameras and three L642V sound cameras.

**Figure 2 animals-13-03538-f002:**
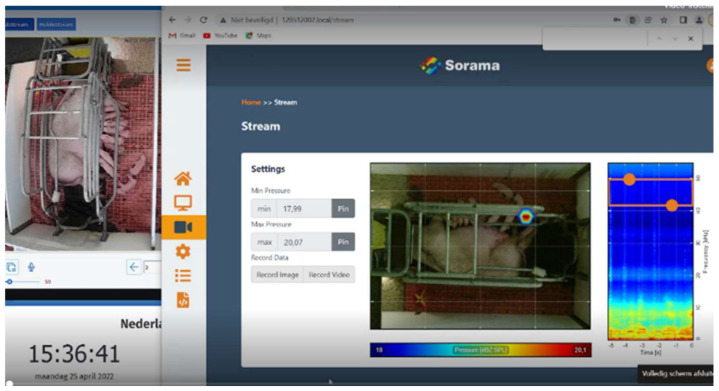
The screen capture that was used for the analysis of behaviour and sound in the farrowing pen.

**Table 1 animals-13-03538-t001:** Sound spots and corresponding audible sounds that were considered correct for field study 1.

Sound Spot	Audible Sound
Head of the sow	Sow in crate
Rump of the sow	Sow in crate
Fence	Metal fence
Head of the sow	Trough
Trough	Trough
Outside pen	Neighbour sow
Cloud of spots	Neighbour sow

**Table 2 animals-13-03538-t002:** Sound spots and corresponding audible sounds and corresponding behaviours that were considered correct for field study 2.

Sound Spot	Audible Sound	Sound Spot	Behaviour
Head of the sow Rump of the sow Fence Head of the sow Trough Outside pen Cloud of spots	Front of the pen Metal fence Metal fence Trough Trough Neighbour sow Neighbour sow	Head of the sow Head of the sow Trough Rump of the sow Fence Fence Outside pen Cloud of spots None	Playing with sack Eating Eating Standing Standing Moving leg lying None None Lying down

**Table 3 animals-13-03538-t003:** Results of visible and audible sounds before farrowing in the first field study (N = 4 sows, 177 min); sound spots without audible sound are false positives (FP), sound spots with audible sound are correct positive (CP); audible sounds without sound spot are false negative (FN) and 2 s of no audible sound and no sound spot indicates a correct negative (CN). If a sound spot was in the wrong location but at the correct time for an audible sound, it was considered either as a false negative (1st column) or as a correct positive (2nd column).

	Wrong Location Considered False Negative	Wrong Location Considered Correct Positive
FP	10,751	10,751
FN	1823	76
CP	1509	3256
CN	1582	1582
Accuracy	19.7	30.9
Error%	80.3	69.1
Sensitivity	45.3	97.7
Specificity	12.8	12.8

**Table 4 animals-13-03538-t004:** Sound spots compared with audible sound, visible behaviour, or sound and behaviour combined.

	Sound	Behaviour	Sound and Behaviour
# Sows	4	2	2
Accuracy	50.6	71.4	91.2
Error%	49.4	28.6	8.8
% Correct positive	35.4	57.5	71.3
Precision	41.8	66.8	89.2
Sensitivity	99.7	99.9	99.6
Specificity	23.6	32.7	69.7

**Table 5 animals-13-03538-t005:** Behaviour of sows (N = 4) four hours before farrowing, during farrowing and four hours after farrowing: percentage of time inactive, either lying or sitting/standing, or being active, playing with the jute sack or rooting on the floor.

Behaviours	Before Farrowing	During Farrowing	After Farrowing
Lying	71.7%	93.5%	100.0%
Sitting/standing	3.3%	6.2%	0%
Jute sack/rooting floor	25%	0.3%	0%

## Data Availability

Data in this study are not publicly archived due to privacy reasons since the study was performed at a commercial farm. Datasets are available from the author upon request.

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
