# Peer review of "Using Sound Location to Monitor Farrowing in Sows"

_animals, 2023, doi:10.3390/ani13223538_

Round 1
Reviewer 1 Report
Comments and Suggestions for Authors
The paper deals with an interesting topic on the use of PLF to create tools to support farmer management and enhance animal welfare. It seems to the reviewer and this is somewhat acknowledged by the authors that the study is a providing preliminary results, promising ones to keep developing the new Technology (since the results presented are based on a limited number of sows evaluated and contexts/scenarios).
There are a few aspects that could be explained a bit in more detail to improve the understanding of the paper or to consider for the discussion.
Specific comments/spelling mistakes:
-Line 77, should be probably slatted floor and not stated
-Line 102, nest-building
More general comments:
- The reviewer is not totally sure if Animals will ask the authors to acknowledge that since the experiment did not involve any management of the sows, but just recording of sound and behaviour of standard practice in a commercial farm, no ethical permit was required. Maybe you will be asked to answer that or include a sentence in the paper.
-Introduction: given the final/long term objectives of the sound camera implementation (prevention of piglet crushing), the introduction is quite long but focuses basically in stillbirths and not in piglet crushing after birth. There is a complete introduction of sow behaviour around farrowing, but very little on the causes of piglet crushing (to a good extent due to postural changes).
-The Observer by NOldus is mentioned in the abstract as a validation means for the sound camera, but then in material and methods, it is no longer mentioned. It is not totally clear how the validation was conducted. Was it a human observer using The Observer to score the false positives/negatives from the security camera recordings? You explain this in the discussion, but it would be good to provide more details in the discussion on how the validation was done.
-Similarly, even though the formulas to calculate “specificity”, “accuracy”, “sensitivity”, “error” are quite well known, it would be good to define them in your material and methods.
-Line 144/145 three behaviours is a quite simple ethogram for piglet crushing. Sitting/standing is considered an inactive behaviour, like lying. But it is actually the change from position to another which causes most of piglet crushing. So, probably, in further steps, when you validate the system for piglet crushing, you will need to include other behaviours and see how well they correlate with sound. It would be good to include postural change, since this is the behaviour basically causing most of piglet crushing (postural change can be different lying positions, from lying to standing and viceversa…).
-Table 3, is the 76% false negatives when wrong location is considered as a correct positive correct? Do false negatives increase/change from 1,8 to 76%?
-Line 204, the reviewer probably does not understand properly the materials. When you say that when comparing sound spot with behaviour and sound, this increases the accuracy, do you mean Behaviour and sound (the two of them happening at the same time in the safety camera, and correlating with a sound spot) or Behaviour or sound (this is either one or the other happening at the time of the sound spot)?
-Line 213-214, why do you mention drinking of the piglets with regards to the rear part of the sow just after birth? Were you expecting drinking at that time, or do you mean suckling?
Discussion/Overall:
-Although the system is aiming at finally preventing piglet crushing, you have done most of your observations and recordings around parturition. It is understood that the field studies will be extended to up to 3 days after farrowing that it is the period when most piglet crushing appears?
-The reviewer is not an expert in pig sounds/calls. However they are usually classified in low-pitch and high-pitch ones, and sometimes associated with different behavioural patterns. Maybe in your study low-pitch sounds would be expected during normal labour of the sow, whereas high-pitch sounds could be more related to piglets being crushed (if they can vocalize) or pain? Is the sound camera able to distinguish those different type of sounds?
-Have you considered to correlate sound with for example facial expression in the sow (also possible to record with a camera), to perhaps identify better when farrowing starts and also the delivery time?
-You have used 5 sows, and individual variability in pain sensitivity, behaviour, maternal abilities (crushing propensity) could be expected. Do you think your results/accuracy could improve if you recorded a higher number of sows/contexts? Probably this will be needed for validation in the future.
Reviewer 2 Report
Comments and Suggestions for Authors
A really interesting study with novel use of microphones to detect issues with farrowing pigs. Interesting to see how much they rooted before farrowing
Summary and abstract make sense
L75 check referencing should be [19-21], also consider word potentially instead of possibly
L83 not sure stagnating the right word
L102 should be nest-building not nets
L199 check spacing
Referencing in the discussion is not numbered as required, make consistant
Otherwise an excellent manuscript with minimal proof reading checks to be done
Do need to explain why there is no ethics number or why not applicable
Comments on the Quality of English LanguageFine, minor proof reading needed
Reviewer 3 Report
Comments and Suggestions for Authors
This study is of significance to sow delivery monitoring and piglet compression detection. But there are still some problems.
(1)The number of experimental animals is insufficient. Only two Bascom XD10-4 security cameras and three Sorama L642V sound cameras were placed.
(2)The algorithm for locating sound sources is not described in detail.
(3)How to avoid noise interference between each sow? How to de-noise sound? None of these issues are clearly articulated.
Round 2
Reviewer 1 Report
Comments and Suggestions for Authors
Dear editors
in my opinion, the authors have addressed my comments and the paper needs to be understood as an initial validation of the method. Therefore the low number of sows involved would be justified provided that the authors already mention in the paper that further research is required.
Author Response
Dear reviewer,
Thank you for reviewing the resubmission of our manuscript and providing valuable feedback. We have shortened the paper to a Communication. We added L238-242 and L319 to the Discussion, and L330-331 to the Conclusions, stating that more research is needed, with more sows and more repetitions, to validate the assumptions.